# SELF-RESOURCE ALLOCATION IN MULTI-AGENT LLM SYSTEMS

## ABSTRACT

With the development of LLMs as agents, there is a growing interest in connecting agents into multi-agent systems to solve tasks concurrently, focusing on their role in task assignment and coordination. This paper explores how LLMs can allocate computational tasks among agent networks, considering factors such as cost, efficiency, and performance. We address key questions, including the effectiveness of LLMs as orchestrators and planners, comparing their efficiency in task assignment and coordination. Our experiments show that LLMs can achieve high validity and accuracy in resource allocation tasks. We find that the planner method outperforms the orchestrator method in handling concurrent actions, resulting in improved efficiency and better utilization of agents. Furthermore, we show that providing explicit information about worker capabilities improves the allocation strategies of planners, particularly when dealing with suboptimal workers.

## 1 INTRODUCTION

Large Language Models (LLMs) have become popular for various tasks beyond just generating text. They can act as agents that interact with their environment (Xi et al., 2025a) and use tools effectively. This has led to the development of more versatile systems that can operate a computer like a human (Agashe et al., 2024; Wu et al., 2024), serve as research assistants (Schmidgall et al., 2025), or even automate tasks in a lab (M. Bran et al., 2024).

As agents are used for more tasks and in more scenarios, they need to interact with other agents. This can happen by design in Multi-Agent Systems (MAS) or through more spontaneous interactions. As a result, frameworks like AutoGen (Wu et al., 2023) and Camel-AI (Li et al., 2023a) have been developed to leverage the capabilities of MAS. The creation of these multi-agent systems opens up new possibilities and challenges to explore (Han et al., 2024; Guo et al., 2024a; Agashe et al., 2023), such as their organizational structures, shared memory, efficiency, and coordination capabilities.

In particular, we focus on how these agents coordinate to assign tasks and achieve their goals. Inspired by Marvin Minsky's idea that intelligence emerges from computational modules working together to accomplish goals that none could achieve alone (Minsky, 1988), we aim to analyze how LLMs allocate tasks to agents. Our goal is to optimize the allocation of resources and tasks by LLMs themselves. Thus, our underlying research question is: **How does a network of LLM-based agents optimize their task allocation?**

We conduct a series of experiments to understand how LLMs allocate tasks to agents within multi-agent systems. First, we evaluate how one LLM can generate the correct task allocation, assigning actions to each agent for a problem with a known solution provided by the Hungarian Algorithm. Second, we examine two methods illustrated in Figure 1—Orchestrator and Planner—using the CuisineWorld benchmark (Gong et al., 2023), detailed in Section 4.2. The Orchestrator uses one LLM to generate all the actions to be executed, while the Planner creates a plan that is then given to executor LLM agents, who generate their actions independently. The plan is re-evaluated when a relevant event occurs. Lastly, we evaluate the Planner allocation based on the agent's abilities.

From the experiments, we concluded interesting findings: First, LLMs achieve higher validity and accuracy in resource allocation tasks as their parameter size increases, but at a significant computational and monetary cost. Second, the planner method outperforms the orchestrator method in handling concurrent actions, resulting in improved efficiency in multi-agent task execution. The planner method achieves better utilization of agents, with fewer idle actions. Finally, we see that LLMs are sensitive to worker capabilities and struggle to infer them dynamically. Providing explicit information about worker capabilities improves planner's allocation strategies, especially with suboptimal workers.

Figure 1: MultiAgents Systems on CuisineWorld (Gong et al., 2023) over 3 methods analyzed: (1) **Individual**: Decentralized, (2) **Orchestrator**: Centralized, (3) **Planner**: A balanced combination. It generates a plan every few steps which is then given to worker agents to generate their actions.

We highlight the following key contributions from this paper:

- An evaluation of LLMs as effective orchestrators for optimizing task allocation in multi-agent systems by comparing their task allocations against the Hungarian Algorithm optimal solution.

- We demonstrate the differences between the orchestrator and the planner methods when handling concurrent actions, highlighting the efficiency gains achieved by the planner.

- An analysis of how the Planner allocates tasks based on agents' abilities within the system, demonstrating that providing explicit information about worker capabilities enhances allocation strategies, particularly when dealing with suboptimal workers.

As multi-agent systems evolve, optimizing task allocation will be critical for enhancing efficiency and performance across applications. This work addresses current challenges in LLM-based frameworks and paves the way for advancements in autonomous and collaborative AI systems.

## 2 RELATED WORK

**Agent Interaction and Collaboration** As more agents are deployed in real-world settings, studying their interactions becomes increasingly relevant. These agents need to communicate not only with each other but also with humans (Jiang et al., 2025; Liu et al., 2023a). Agents can choose to cooperate, compete, or do a mix of both (Tran et al., 2025). Since these models use natural language for communication, recent studies have explored the concept of "Theory of Min", which involves understanding and attributing *mental* states to oneself and others (Strachan et al., 2024; Street, 2024). Some works apply Game Theory to analyze these interactions (Hua et al., 2024). Several studies measure the coordination abilities of LLMs, such as the LLM-Coordination Benchmark (Agashe et al., 2023) and MindAgent (Gong et al., 2023), Gamma($\gamma$)-Bench (Huang et al., 2024). Other works focus on improving the abilities of LLMs to interact and coordinate (Li et al., 2023b; Zhang et al., 2024a; 2023b; Cross et al., 2024; Zhang et al., 2023a; Guo et al., 2024b).

**LLM-based Multi-Agent Systems** Extending single-agent systems to multi-agent systems has led to increased computational efficiency during inference, with shorter execution times. Multi-agent systems offer several benefits, including modularity, specialization, collaborative learning, and improved decision-making. These systems provide better scalability and flexibility, making them more effective at solving problems that a single model might struggle with. The development of multi-agent systems has led to the creation of various frameworks focused on MAS, such as AutoGen (Wu et al., 2023), Camel-AI (Li et al., 2023a), or MetaGPT (Hong et al., 2023). These frameworks have enabled the creation of systems with better scalability and flexibility, enhancing problem-solving capabilities. For example, the Chain of Agents (Zhang et al., 2024b) can process long contexts effectively, FilmAgent (Xu et al., 2024) generate coherent long-sequence videos , Chemrow (M. Bran et al., 2024) support new chemistry research, and Smart-LLM to control multiple robots simultaneously (Kannan et al., 2024).

Table 1: Comparison of the three experimental scenarios

|  | Experiment 1 | Experiment 2 | Experiment 3 |
| --- | --- | --- | --- |
| Objective | Minimize total cost | Maximize completed tasks | Maximize completed tasks |
| Environment | Static assignment problem | Dynamic CuisineWorld | Dynamic CuisineWorld with varying agent capabilities |
| Reward Structure | Explicit costs | Delayed rewards | Delayed rewards |
| Agent Capabilities | Uniform | Uniform | Varied |
| Decision Making | Centralized | Centralized (Orchestrator) vs. Semi-Decentralized (Planner) | Semi-Decentralized (Planner) with capability awareness |
| Evaluation Metric | Accuracy and Validity rate | Task Completion Rate and Efficiency | Task Completion Rate and Efficiency |

**Learning to optimize resources**  When developing multi-agent systems, a crucial question arises: What is the optimal architecture? This is explored in Zhuge et al. (2024); Liu et al. (2023b). Once the architecture is defined, how do these systems utilize it effectively? First, selecting the right model for specific queries can optimize performance by ensuring high-quality answers, as explored in RouteLLM (Ong et al., 2024) Hybrid-LLM (Ding et al., 2024), and the corresponding benchmark RouterEval (Huang et al., 2025). In Mindstorms (Zhuge et al., 2023), the authors theorize how LLMs can lead to systems that self-manage resources based on a monetary system, optimizing resources and maximizing rewards to create an "*Economy of Minds*" (EOM). Models can also learn to optimize their communication, as demonstrated in Agora (Marro et al., 2024). Self-organized MASs can reduce the burden on developers while achieving better results, such as in code generation (Ishibashi & Nishimura, 2024) or leveraging scale in multi-agent reward-based learning (Slumbers et al., 2023; Ma et al., 2024; Tekin et al., 2025).

## 3  PROBLEM DEFINITION

Large language models are increasingly deployed as agents within multi-agent systems, where they must coordinate to assign and execute tasks. Our focus is on how these agents organize themselves to become more autonomous and optimize task allocation under two key criteria: **cost** and **performance**. Cost is measured in terms of token usage and model size, while performance reflects the accuracy and efficiency of completing the tasks. Formally, given a task $T$ and a set of agents $\mathcal{A} = \{a_1, \ldots, a_N\}$, we aim to identify an organizational strategy $\Omega$ that balances these objectives:

$$\Omega^* = \arg\max_{\Omega} \text{Performance}(T, \Omega) - \lambda \cdot \text{Cost}(T, \Omega), \tag{1}$$

where $\lambda$ is a tradeoff parameter between performance and cost.

**Agents**  Each agent $a_i \in \mathcal{A}$ is defined by:

- **Cost** ($c_i$)**:** computational expense of using the agent, based on token usage and parameter size.
- **Capability** ($\phi_i$)**:** the proficiency of the agent in performing tasks of varying difficulty.

**Tasks**  A task $t_j \in \mathcal{T}$ may vary in complexity and can be decomposed into subtasks. Different agents may perform differently depending on the subtask assigned, both in terms of cost and performance.

**Organizational Strategies**  We study three approaches to structuring multi-agent LLM systems:

1. **Decentralized:** All agents act independently without central coordination.
2. **Centralized Orchestrator:** A central agent assigns concrete actions to simpler executor agents.
3. **Centralized Planner:** A central agent decomposes the task into subtasks, while executor agents reason independently about how to achieve their assigned subtask.

# 4 EXPERIMENTAL FRAMEWORK

## 4.1 EXPERIMENT 1: BASIC RESOURCE ALLOCATION

**Problem Definition** Orchestrator agents are widely adopted in multi-agent systems to allocate resources efficiently. We model the orchestrator's objective as a resource allocation problem where a set of agents must complete a sequence of tasks. While long-horizon tasks typically involve dynamic environmental updates (as shown in Figure 2), we focus here on the fundamental single-turn assignment problem. By analyzing this problem, we focus on the ability of LLMs to allocate resources in a setting where costs and rewards are explicitly defined.

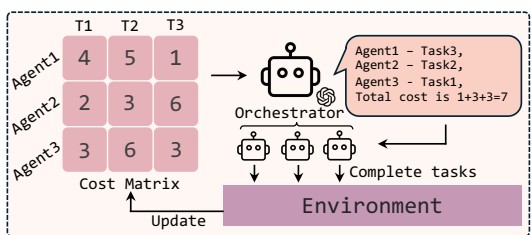

Figure 2: A multi-agent system with an LLM-based orchestrator. At each step, the orchestrator assigns multiple tasks to agents and minimizes the total cost. After agents complete their tasks, the environment updates with new tasks and a new cost matrix, continuing the process iteratively.

Consider a system comprising a set of $N$ agents $\mathcal{A} = \{a_1, \ldots, a_N\}$ and a set of $P$ tasks $\mathcal{T} = \{t_1, \ldots, t_P\}$. Each agent $a_i$ is characterized by an operational cost (computational expense) and a capability measure $\phi_i$ (model proficiency). Tasks may differ in difficulty $d_p$ and workload $w_p$, and each task $t_p$ can be decomposed into a sequence of $M_p$ subtasks. The utility of assigning agent $i$ to subtask $m$ of task $p$ is defined as $u_{pim} = q_{pim} - c_{pim}$, representing the quality of output minus the execution cost. If an agent cannot execute a specific subtask, the utility is $-\infty$.

The orchestrator aims to maximize the total system utility by determining the binary assignment variables $v_{pim} \in \{0, 1\}$. The optimization is constrained by a maximum time budget $T_{max}$ and the requirement that each subtask is assigned to at most one agent:

$$\arg\max_v \sum_{p=1}^{P} \sum_{i=1}^{N} \sum_{m=1}^{M_p} u_{pim} v_{pim} \tag{2}$$

Subject to:

$$\sum_p \sum_i \sum_m \tau_{pim} v_{pim} \leq T_{max} \tag{3}$$

$$\sum_i v_{pim} \leq 1 \quad \forall m \in \mathcal{M}_p, \forall p \in \mathcal{P} \tag{4}$$

where $\tau_{pim}$ denotes the time required for the assignment. As this general formulation is NP-hard (Korsah et al., 2013; Gong et al., 2023), we employ Large Language Models to approximate the solution.

To rigorously evaluate the LLM orchestrator's baseline capability, we first simplify this general framework into the standard *Linear Assignment Problem*. In this setting (Experiment 1), we align the number of agents and tasks ($N = P$) and treat tasks as atomic units ($M_p = 1$). We focus solely on cost minimization by setting quality $q = 0$ and time $\tau = 1$, such that $T_{max} = N$ allows for a one-to-one mapping. The objective simplifies to finding the optimal assignment matrix $v$ given a cost matrix $A \in \mathbb{R}^{N \times N}$:

$$\arg\min_v \sum_{i=1}^{N} \sum_{j=1}^{N} A_{ij} v_{ij} \tag{5}$$

subject to $\sum_i v_{ij} = 1$ and $\sum_j v_{ij} = 1$. This reduction allows us to benchmark the LLM against the optimal polynomial-time Hungarian algorithm ($O(N^3)$), verifying whether the orchestrator can effectively minimize costs "out of the box" before addressing more complex multi-agent scenarios.

**Experiment Details**   To evaluate the LLM performance on assignment problems, we first created an evaluation set of 100 samples by generating 6x6 matrices with random numbers. After that, we utilize the Hungarian algorithm to get the ground truth for each answer. All LLM orchestrators are evaluated using greedy decoding with a maximum sequence length of 2048. By standardizing the dataset and decoding method, we aim to isolate the contribution of the model architecture itself. This allows us to better understand which LLMs naturally exhibit stronger combinatorial reasoning skills.

**Evaluation**   We evaluate the performance of self-resource allocation across LLMs of varying model sizes and model providers. Specifically, we consider GPT-4o-mini(∼28B), Mistral-Small-3.1(24B), Qwen2.5-32B-Instruct, Llama-3.1-70B-Instruct, GPT-4o(∼200B), and Llama3.1-405B-Instruct-FP8. Two evaluation metrics, accuracy and validity rate, are used in the results: ❶ **Accuracy** measures how many of the LLM-generated assignment solutions are optimal. A solution is considered as correct only when it is exactly the same as the ground truth optimal assignment given by the Hungarian algorithm. Given the variability of responses and the difficulty to parse them correctly, we use the LLM judge based on GPT-4o to compare each model response to the ground truth and define it as *correct* or *incorrect*. ❷ **Validity rate** tests how many of the LLM-generated assignment solutions are valid, which can be not optimal. Invalid assignments include assigning one agent to more than one task, leaving some tasks not assigned with agents, and making up an incorrect lower cost of the agent.

## 4.2   EXPERIMENT 2: CONCURRENT ALLOCATION

**Problem Statement**   In this experiment, we introduce additional complexity by incorporating delayed rewards, as outlined in Table 1. The reward is provided only after a sequence of actions has been completed. This allows to simulate a case closer to real-world scenarios, where agents often need to execute actions without immediate feedback, and they may not know in advance whether those actions will lead to a positive outcome.

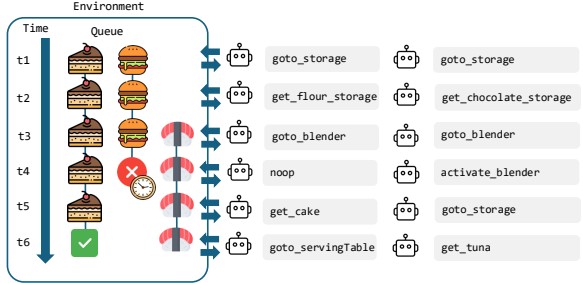

To model this behavior, we use *CuisineWorld* (Gong et al., 2023) as the benchmark, inspired by Overcooked! (Carroll et al., 2019). In this environment,

Figure 3: In CuisineWorld, the agents are presented with a set of dishes to complete in certain. Agents execute tasks until dishes are completed or the order expires.

agents must complete dish orders by collecting and cooking ingredients, then delivering them within a time frame. The goal is to maximize the amount of completed orders. The benchmark features 10 locations, 27 ingredient types, and 33 dishes, categorized by difficulty and cooking tools required, resulting in 12 game levels. Figure 3 represents this game setting, where dish orders appear at regular intervals and each order has a predefined time limit for completion. Each dish also requires multiple steps to be executed before it can be completed. We refer to Appendix B for more information.

As originally described, the agent decision process can be formulated as a *Markov Decision Process* $(\mathcal{S}, \mathcal{A}, \mathcal{T}, \mathcal{M}, \mathcal{G})$ with state space $\mathcal{S}$, action space $\mathcal{A}$, (indicating all the possible schedules that can be made at a single time step), transition dynamics $\mathcal{T}$, reward function $\mathcal{R}$ and task instruction space $\mathcal{G}$.

• State Space $\mathcal{S}$: The environment consists of two main entities: locations and agents. Locations can be storage areas, serving tables, or cooking tools like pans and blenders. Each location's description includes the items it contains and whether it is currently occupied. Agents are described by their current location, the items they are holding, and whether they are using a tool.

• Actions: Actions in CuisineWorld involve dispatching commands to agents, such as moving to a location, obtaining or placing items, activating tools, or performing no operation. The list of actions include `goto` (agent, location), `get` (agent, location, item), `put` (agent, location), `activate` (agent, location) and the idle action.

• Tasks: The tasks involve completing dish orders, which range from simple to complex recipes. New tasks are introduced at regular intervals, and each task has a limited time frame within which it must be completed, otherwise it fails.

**Experiment Details** The goal of this experiment is to evaluate the effectiveness of planning in a multi-agent LLMs compared to the use of an orchestrator. We follow the definition of an agent provided in React (Yao et al., 2023) and (Xi et al., 2025b), where agents are defined as: *environment* $\rightarrow$ *reasoning* $\rightarrow$ *action*. We compare three methods, represented in Figure 1 from $n = 1, ..., 6$ agents:

1. *Individual*: Every agent is controlled by a different LLM, and generates its own action independently.

2. *Orchestrator*: One LLM controls all the agents in the game and generates actions for all of them.

3. *Planner*: Generates a general plan whenever a relevant event occurs in the game, such as when a dish is introduced, completed, or removed. This plan is then provided to the LLM agents, which produce their independent actions.

**Evaluation** We evaluate the performance of the previous methods in producing the correct allocation and maximizing the number of orders completed. We aim to study how individual, independent weak models (decentralized) can benefit from planning by a larger model. For this purpose, we use Llama 70B-Instruct, Qwen 32B-Instruct, and GPT4o-mini as executor models, while GPT-4o and Claude 3.7 Sonnet are used for the Orchestrator (centralized). The Planner method is then evaluated using a combination of these models, where GPT-4o and Claude 3.7 generate the plans, and open-source models generate the actions. To evaluate the results, we use the following metrics: ❶ **#Completed Orders**: the number of orders the agents are able to complete; ❷ **Execution Cost**: the cost of LLM calls at current prices, detailed in Table 6, which closely represents their running cost; and ❸ **Efficiency**=$\frac{\#complted\_orders}{\$cost}$: the ratio of completed orders to cost, representing the work done per dollar spent.

### 4.3 EXPERIMENT 3: CAPABILITY-AWARE ALLOCATION

**Problem Statement.** In the third experiment, we want to evaluate the capability of LLMs to allocate tasks based on the ability of the worker agents. We fix the Planner model and evaluate its ability to allocate plans to a heterogeneous mix of worker agents, each with a different backbone model. Moreover, each of these backbone models has varying levels of intelligence and parameter sizes, allowing us to evaluate the task distribution of LLMs to Worker LLMs of varying intelligence capabilities. Capability-aware allocation introduces several challenges compared to the previous settings. First, the planner must reason about differing skill levels, which increases the complexity of the allocation strategy. Second, the planner must correctly infer which workers are best suited for some subtasks. These challenges make this experiment a stronger test of planning ability than previous ones.

**Experiment Details** We use the same CuisineWorld environment Gong et al. (2023) - with similar settings as Experiment 2 (Section 4.2), but different set of worker models. We run two experimental settings to evaluate the Capability-Aware Allocation abilities of LLMs as Planners. In the first setting, *On-the-fly Allocation*, the Planner works similarly to the previous cases, where the only signals available to the model for inferring worker capability are those implicitly contained in the evolving environment trajectory (whether actions succeed or whether progress is made on a dish). The Planner has to decide on the possible plan based on its own reasoning over these observed states. In the second setting, *Informed Allocation*, the Planner is provided with lightweight capability descriptors for each worker agent ($\phi_i$), derived from their individual action-success rates in CuisineWorld from the previous experiments. The action success rates of the individual models on CuisineWorld is a proxy measure of their capability in the environment. Together, these two settings allow us to quantify (1) how robustly a Planner can adapt to heterogeneous teams without prior calibration, and (2) results from even minimal capability information.

**Evaluation:** For evaluation we use the Claude-3.7-sonnet model as the planner, as it shows to have the best planning capabilities. We vary the heterogeneous worker models from the pool of: Llama-3.1-70B-Instruct, Qwen2.5-32B-Instruct, and GPT-4o-mini. We run a total of 7 experiments with varying combinations of these models for $n = 1, 2, 3$ worker agents. To evaluate the results, we use ❶ **Efficiency** ($= \frac{\#complted\_orders}{\$cost}$) to measure the effectiveness of the task allocation strategies in terms of the number of orders completed relative to the cost incurred.

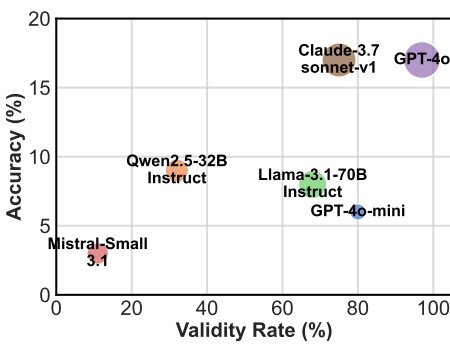

Figure 4: Performance of LLM orchestrators on assignment problems. The circle size scales with the model parameter count.

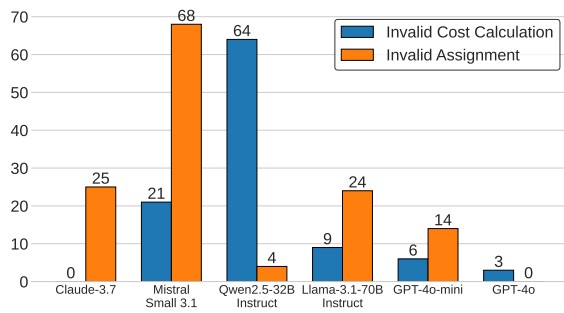

Figure 5: Experimental Error Analysis. It shows the LLM errors in the assignment spitted by "invalid cost calculation" (blue) and "invalid assignment" (orange).

## 5 RESULTS

**LLMs address the assignment problem, with performance scaling alongside model size and cost.**
As shown in the Figure 4, larger models generally achieve higher validity and accuracy in resource allocation tasks, indicating that orchestration performance improves with model capacity. However, these gains come at a significant computational and monetary cost: the most capable models are also the most expensive to deploy. Consequently, while an advanced LLM offers superior orchestration abilities, thus raising the need for an alternative more efficient solution for the resource allocation in multi-agent systems.

*Error Analysis* Looking at the results, we identify that invalid assignments arise from two main sources: (1) *Invalid cost calculation*: the reported solution cost is lower than the minimum feasible cost (e.g., the orchestrator provides a valid assignment but incorrectly reports a lower total cost, or makes up a lower cost of assigning a specific agent to a task). (2) *Invalid assignment*: an agent is assigned to multiple tasks, or a single task is assigned to multiple agents. We record the number of invalid assignments for every model under each cause in Figure 5 and include an example in Appendix A. These observations suggest that improving the reliability of cost estimation and enforcing assignment constraints are key directions for reducing error rates in future systems.

**The Planner method achieves better efficiency.** The results are presented in Table 2, where we see that central planning completes the highest number of orders, as the model counts with all the information to take decisions with a strong model. These results are not surprising given that we are using a strong model at every step, thus inquiring in a higher overall cost. However, the results are different when measuring the efficiency of the MAS. As described previously, we refer to efficiency as the ratio between the number of orders and the cost. In Figure 6, we plot the efficiency results for all models. This shows how the Planner method becomes the most cost-efficient for the results obtained.

Furthermore, in Figure 7, we see how the Planner method achieves better utilization of the agents, with a lower percentage of idle actions generated. As the agents are given more independence to generate their actions, they tend to produce fewer idle steps than when centrally organized.

**LLMs struggle to infer worker capabilities but improve when given explicit hints.** Figure 8 compares On-the-fly Allocation to Informed Allocation. In On-the-fly allocation, without any prior indication of each worker LLM's strengths, planners struggle to identify the best allocations dynamically, especially when worker performance varies widely. However, simply providing subtle cues about worker capabilities (action success rate from On-the-fly allocation performance) in Informed Allocation leads to a marked increase in overall efficiency—particularly when planners must work with suboptimal models. These cues reduce non-productive actions, helping LLM-based planners better match tasks to models when aware of worker differences.

Table 2: Results from Experiment 2 (Section 4.2): Concurrent Allocation in CuisineWorld the number of completed orders and associated costs for different models and methods

| Models | | Completed Orders | | | | | | Cost ($) | | | | | |
| --- | --- | --- | --- | --- | --- | --- | --- | --- | --- | --- | --- | --- | --- |
| | | #Agents | | | | | | #Agents | | | | | |
| Plan./Orch. | Worker | 1 | 2 | 3 | 4 | 5 | 6 | 1 | 2 | 3 | 4 | 5 | 6 |
| *Individual* | | | | | | | | | | | | | |
| ✗ | GPT-4o-mini | 1 | 3 | 3 | 3 | 4 | 5 | 0.8 | 1.6 | 2.6 | 3.6 | 4.6 | 5.7 |
| ✗ | Llama-70B | 14 | 25 | 22 | 34 | 33 | 40 | 3.7 | 7.7 | 12.0 | 16.4 | 21.1 | 26.0 |
| ✗ | Qwen-32B | 9 | 20 | 20 | 19 | 20 | 26 | 2.0 | 4.3 | 6.7 | 9.0 | 11.5 | 12.0 |
| ✗ | GPT-4o | 24 | 29 | 37 | 44 | 48 | 51 | 11.6 | 24.3 | 37.5 | 51.2 | 65.5 | 80.1 |
| ✗ | Claude-3.7 | 23 | 40 | 46 | 54 | 59 | 59 | 16.4 | 35.2 | 57.2 | 79.8 | 104.4 | 126.6 |
| *Orchestrator* | | | | | | | | | | | | | |
| GPT-4o | ✗ | 20 | 37 | 33 | 48 | 34 | 40 | 11.6 | 12.6 | 13.6 | 14.2 | 15.0 | 15.8 |
| Claude 3.7 | ✗ | 26 | 49 | 66 | 85 | 85 | 98 | 17.5 | 21.0 | 22.7 | 24.1 | 25.9 | 27.1 |
| *Planner* | | | | | | | | | | | | | |
| GPT-4o | GPT-4o-mini | 9 | 11 | 12 | 13 | 12 | 11 | 2.3 | 2.8 | 2.7 | 2.6 | 2.2 | 2.1 |
| | Llama70B | 21 | 27 | 41 | 40 | 48 | 42 | 4.4 | 6.5 | 8.6 | 10.5 | 12.8 | 15.0 |
| | Qwen32B | 11 | 22 | 22 | 24 | 24 | 22 | 3.8 | 4.7 | 5.7 | 6.9 | 7.8 | 9.0 |
| Claude-3.7 | GPT-4o-mini | 3 | 9 | 17 | 20 | 22 | 30 | 1.5 | 4.4 | 5.0 | 5.5 | 6.3 | 6.8 |
| | Llama-70B | 21 | 44 | 57 | 68 | 72 | 77 | 5.1 | 7.2 | 9.4 | 11.4 | 13.7 | 15.9 |
| | Qwen32B | 16 | 30 | 42 | 43 | 50 | 50 | 4.7 | 5.5 | 6.7 | 7.8 | 9.1 | 10.1 |

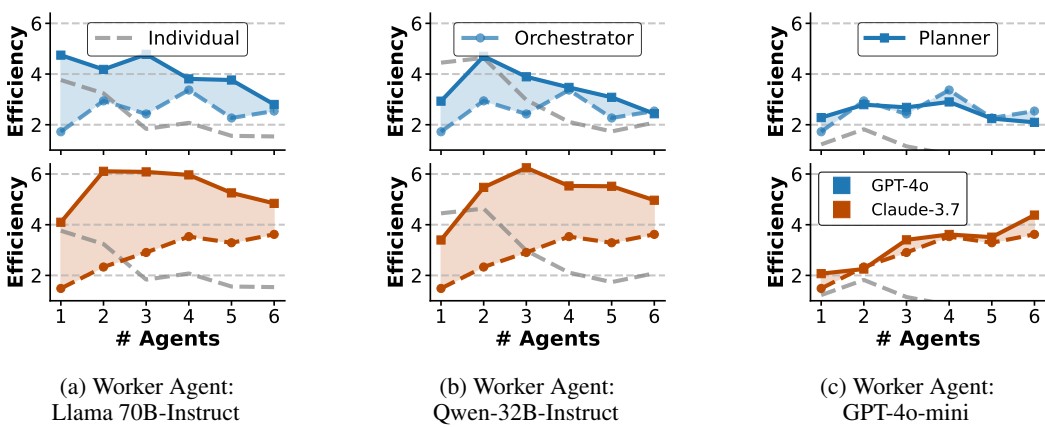

(a) Worker Agent:
Llama 70B-Instruct

(b) Worker Agent:
Qwen-32B-Instruct

(c) Worker Agent:
GPT-4o-mini

Figure 6: Efficiency for Planner, Orchestrator and Individual methods. Top (blue) plots use GPT-4o as orchestrator/planner. Bottom plots (brown) use Clause-3.7 as orchestrator/planner. The results demonstrate the planner method is more efficient than their centralized or decentralized counterparts.

**LLMs are highly sensitive to worker capabilities.** We find the performance of a multi-agent system is significantly influenced by the specific capabilities of each worker LLM and how these capabilities are combined. In a homogeneous setting, Llama-70B-Instruct and Qwen32B consistently outperform GPT-4o-mini when used as worker models. Mixing models of varying strengths generally reduces average efficiency. However, including at least one stronger model in a heterogeneous team improves efficiency. For instance, combining Qwen with GPT-4o-mini yields an efficiency of 4.04, compared to 2.79 for two GPT-4o-mini models from experiment 2. In the heterogenous case, we observe that a smaller team of more capable LLMs can outperform a larger team with uneven skills, as seen with the Llama–Qwen pairing outperforming the combination of all three.

**Ablation Study. Measuring the effect of prompts.** Finally, we also run an ablation study on the effect of the prompt to the results. To quantify how sensitive our conclusions are to wording choices, we generated three semantically-equivalent paraphrases of each template block, included in Appendix F. We reran the best-performing models and confirmed that the main efficiency results hold (Table 9), though the Planner case showed greater variability due to the use of GPT-4o-mini. Specific data is include in Appendix Appendix I.

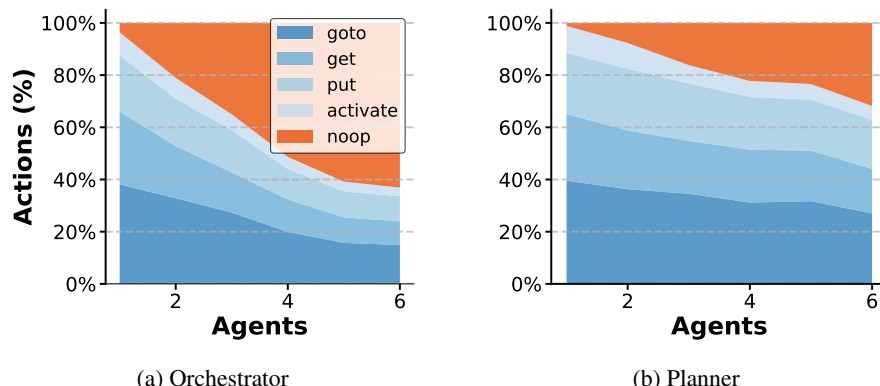

(a) Orchestrator          (b) Planner

Figure 7: Percentage of actions executed by action type in Experiment 2 (*Orchestrator = GPT-4o; Planner = GPT4o+Llama70B-Instruct*). The results indicate that planners maintain a higher percentage of active (non-idle) actions compared to a centralized orchestrator.

Table 3: Ablation Study. Efficiency results for the paraphrases prompts. It presents the results for Planner (Claude 3.7) and Planner (Claude 3.7+GPT-4o-mini), and the difference ($\Delta$) with the original experiments. Results preserve overall efficiency trends, with higher variability in the Planner case.

| #Agents | 1 | 2 | 3 | 4 | 5 | 6 | Avg. |
|---|---|---|---|---|---|---|---|
| **Orchestrator (Claude-3.7)** | 1.49 | 2.33 | 2.91 | 3.53 | 3.29 | 2.62 | |
| $\Delta$(Original-Ablation) | +0.06 | +0.04 | +0.56 | –0.11 | +0.13 | –0.33 | 0.21 |
| **Planner (+ GPT-4o-mini)** | 2.07 | 2.05 | 3.41 | 3.62 | 3.51 | 4.38 | |
| $\Delta$(Original-Ablation) | +0.38 | +2.43 | +1.07 | +1.50 | –0.03 | –1.09 | 1.07 |

## 6 CONCLUSIONS

This work explores the capabilities of LLMs to optimize task allocations in multi-agents systems. Experiments show that LLMs can function as orchestrators, with relative performance to that of established algorithms like the Hungarian Algorithm. However, using a planner method instead of an orchestrator improves efficiency in handling concurrent actions. These findings suggest that leveraging LLMs can create more efficient and cost-effective multi-agent frameworks, dynamically allocating resources based on real-time needs, enhancing performance and cost-effectiveness.

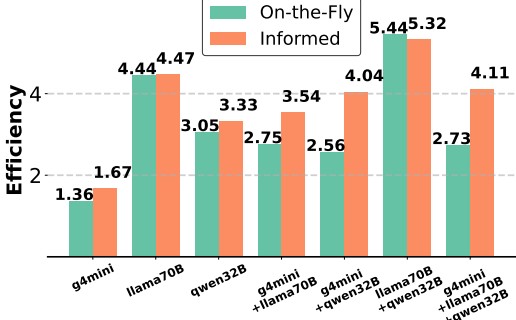

Figure 8: Efficiency comparison of On-the-fly Allocation and Informed Allocation for Capability-Aware Task Allocation with LLMs as Planners.

## ETHICS STATEMENT

This paper aims to automate processes using AI systems, focusing on developing efficient and intelligent multi-agent frameworks for applications like research assistance and task management. We acknowledge the societal implications of our work and emphasize the need for further analysis to understand LLM outputs, especially in real-world applications. We advocate for transparent algorithmic processes that allow individuals to comprehend AI-driven decisions. Our experiments are conducted in a controlled gaming environment to ensure safety and prevent unintended impacts on other systems.

## REPRODUCIBILITY STATEMENT

We have made efforts to ensure the reproducibility of our results. The problem formulation and optimization setup are detailed in Section 3. The three experimental settings—including the assignment problem, the CuisineWorld concurrent allocation, and the capability-aware allocation—are described in Section 4, with further details of the CuisineWorld environment in Appendix B. Model configurations, costs, and training/evaluation protocols are reported in Section 5, with pricing and ablation studies given in Appendices C–E. All prompts used for orchestration, planning, execution, and ablations are provided in Appendices F–I to enable replication of our experiments. Additionally, we commit to make code and results publicly available and contribute to the open-source community.

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

## A  EXPERIMENT 1: EXAMPLE INVALID COST CALCULATION

In this section, we present an example of an invalid cost calculation from the models.

---

**Example: Invalid Cost Calculation in an Assignment Problem**

**User instruction:** You are an AI assistant who helps to solve assignment problems. You are given an *assignment problem* in which there is a bipartite graph. The left set (e.g., A, B, C...) might represent agents (drivers, employees, etc.), and the right set (e.g., #101, #102, etc.) might represent tasks (packages, jobs, etc.). Each edge weight is the cost or time of assigning that agent to that task. Your task:

1. Read the input bipartite graph (in JSON format).

2. Compute a minimum-cost combination to assign each task to one agent. Each agent can only be assigned to one task.

3. Return the result in a specific JSON output format.

**Input bipartite graph (JSON):**

```
{"A": {"101": 2, "102": 6, "103": 5}, "B": {"101": 7, "102": 4,
    "103": 2}, "C": {"101": 3, "102": 3, "103": 4}}
```

**Orchestrator output:**

```
Assignment: [["A","101",2], ["B","103",2], ["C","103",3]]
Final cost: 7
```

---

**Why invalid.** From the input, the edge cost is $w(C, 103) = 4$ (not 3), so the corrected total is $2 + 2 + 4 = 8 \neq 7$. (Additionally, task 103 is assigned twice while task 102 is unassigned, violating the one-to-one constraint.)

## B  CUISINEWORLD

CuisineWorld (Gong et al., 2023) is a benchmark designed to evaluate the planning and coordination capabilities of multi-agent systems. It simulates a virtual kitchen environment where multiple agents need to collaborate to complete various cooking tasks. actions that can be taken by the LLMs are described in 4. Recipes provide a step-by-step guide for preparing different dishes. They list the required ingredients for each part of the process, the tools you'll need, and what the final dish should look like once it's cooked. The recipes are grouped into 13 levels, with different range of difficulty depending on the number of cooking tools involved, the number of ingredients, and the number of steps required to complete the dish. Figure 9 shows the distribution of these factors over these groups. Data provided from the environment is given to the agents in text form, as shown in Table 5.

Table 4: Action space in CuisineWorld.

| Type | Arguments | Description |
|------|-----------|-------------|
| goto | agent location | Move agent to location |
| get | agent location (item) | agent obtain item from location |
| put | agent location | agent put everything it holds to location |
| activate | agent location | agent turn on location |
| noop | agent location | not dispatching agent |

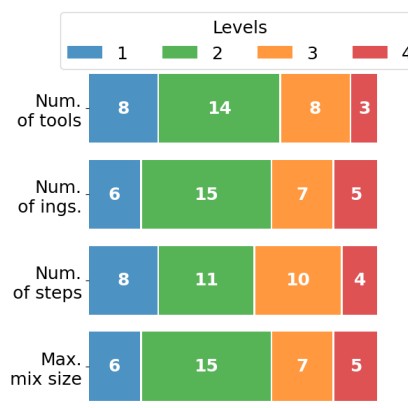

Figure 9: Dish distribution over the number of tools and ingredients (ings.) involved, cooking steps, and maximum mixture size as in the recipe.

Table 5: Example of Game State returned from the environment and provided to the agents.

| **Game Configuration** | |
|---|---|
| Current Game Level | level_1 |
| Current Dishes | |
|    Name | salmonMeatcake |
|    Lifetime | 10 |
| Current Game Step | 10 |
| Maximum Game Steps | 60 |
| **Agent State** | |
| at(agent0, servingtable0) | |
| hold(agent0, None) | |
| at(agent1, servingtable0) | |
| hold(agent1, None) | |
| **Kitchen State** | |
| inside(storage0, None) | |
| inside(servingtable0, None) | |
| inside(blender0, None) | |
| inside(blender1, None) | |
| **Accomplished Tasks** | |
| salmonMeatcake | |

## C PROMPT ARCHIVE

### C.1 MAIN PROMPTS

ORCHESTRATOR

```
In this game, there are {total_num_agents} agents available, so ...ould
    generate the actions for all the {total_num_agents} agents.

When asked for reasoning, you will explain your thought process ...the
    current state, what needs to be done by each agent, and why.
When asked for actions, you will provide the actions for all {
    total_num_agents} agents, one action per line.

Possible action types include:
```

```
- goto_agent[id]_[location]
- get_agent[id]_[item]_[location]
- put_agent[id]_[location]
- activate_agent[id]_[appliance]
- noop_agent[id]

Follow the formats exactly as shown in the examples, as your responses
    will be automatically parsed.
```

## PLANNER

```
You are a strategic planner for a multi-agent kitchen environment. Your
    job is to produce a clear, step-by-step high-level plan that
    coordinates all {total_num_agents} agents to complete all dishes
    efficiently. You must reason about tasks, dependencies, appliances,
    and item locations, and update the plan as the environment changes.

Your output must contain two parts: (1) reasoning and (2) plan. The plan
    should list tasks per agent and specify when to use each appliance or
     tool. The plan must be actionable by executor agents.

You will be provided with the current goal(s), the partial progress, and
     a summary of the kitchen environment:
- The layout of the kitchen and the positions of items
- The statuses of appliances (on/off, in-use)
- Any constraints or bottlenecks

Be concise but complete. Do not invent objects or locations that are not
    in the provided environment summary.
```

## EXECUTOR

```
You are an executor agent (Agent {agent_id}) in a kitchen environment.
    You will be given a high-level plan that describes how all agents
    should work together, as well as the current local observations and
    your inventory.

Your job is to choose the next low-level action for Agent {agent_id} that
     best advances the plan. Consider collisions, current appliance
    states, and item availability. If you cannot make progress, choose a
    safe noop and explain why.

Output two parts:
1) reasoning_agent{agent_id}: <your short reasoning>
2) action_agent{agent_id}: <one atomic action>

Valid action types include:
- goto_agent{agent_id}_[location]
- get_agent{agent_id}_[item]_[location]
- put_agent{agent_id}_[location]
- activate_agent{agent_id}_[appliance]
- noop_agent{agent_id}

Follow the exact output format and only produce the next action in the
    format: action_agent{agent_id}_parameters
```

## D EXAMPLES (SHORTENED)

## E EXAMPLES

### AGENT EXAMPLE

```
User:
----- START OF EXAMPLE -----
2 agents available. You are controlling agent0.
Goal: Make a salmonMeatcake.

-game state:
current dishes: salmonMeatcake (lifetime 10)
agent0 at servingtable0, holding None
agent1 at servingtable0, holding None

Assistant:
I need to make a salmonMeatcake. I'm at servingtable0 holding nothing.
Step 1: Go to storage0 to get salmon, the main ingredient.
```

### ORCHESTRATOR EXAMPLE

```
User:
----- START OF EXAMPLE -----
2 agents available. Both start at servingtable0.

-game state:
current dishes: salmonMeatcake (lifetime 10)
agents not holding anything

Assistant:
Both agents must work together. Agent0 should get salmon from storage0.
Agent1 should get flour from storage0. This prepares both ingredients in
    parallel.
```

### PLANNER EXAMPLE

```
User:
----- START OF EXAMPLE -----
2 agents available. Goal: Make tunaSashimi.

-game state:
current dishes: tunaSashimi (lifetime 10)
agents at servingtable0, holding nothing

Assistant:
Agent0 retrieves raw tuna from storage and brings it to the chopboard.
Agent1 moves to the chopboard, chops the tuna, and delivers sashimi to
    servingtable.
```

## F PROMPTS USED IN ABLATIONS

We report here the distinct prompts used for the ablation conditions. Where 3 prompts different to the main prompts are generated.

### ABLATION: ORCHESTRATOR

```
This game involves {total_num_agents} agents. Your task is to generate
    actions for all {total_num_agents} agents.

If prompted for reasoning, clearly describe your thought process,
    including the current situation, each agent's responsibilities, and
    the rationale behind the plan.

If prompted for actions, list one action per agent, with each on a
    separate line.

Permitted action types are:
- goto_agent[id]_[location]
- get_agent[id]_[item]_[location]
- put_agent[id]_[location]
- activate_agent[id]_[appliance]
- noop_agent[id]

Be precise. Match the formatting shown in the examples exactly, as
    responses are parsed automatically.
```

```
You are participating in a scenario involving {total_num_agents} agents.
    You are responsible for generating an action for each of the {
    total_num_agents} agents.

When reasoning is requested, outline your analysis of the current context,
     what tasks are needed, which agents should perform them, and your
    justification.

When actions are requested, return one action per agent, formatted as one
     action per line.

Supported action types are:
- goto_agent[id]_[location]
- get_agent[id]_[item]_[location]
- put_agent[id]_[location]
- activate_agent[id]_[appliance]
- noop_agent[id]

Adhere strictly to the formats provided above responses are
    programmatically parsed
```

```
There are {total_num_agents} agents in this game. You must output actions
    for each of the {total_num_agents} agents.

If prompted for reasoning, explain the current situation, the goal for
    each agent, and your decision-making process.

If prompted for actions, return exactly one action per agent, listed on
    separate lines.

The allowed action formats are:
- goto_agent[id]_[location]
- get_agent[id]_[item]_[location]
- put_agent[id]_[location]
- activate_agent[id]_[appliance]
- noop_agent[id]

Do not deviate from these formats responses will be parsed by a system
    expecting this exact structure.
```

ABLATION: PLANNER

```
You are the strategic planner for a kitchen staffed by {total_num_agents}
    agents.
Your goal is to generate a high-level plan that enables agents to fulfill
    dish orders efficiently.

You have access to:
- The full kitchen layout, including agents, ingredients, and appliances
- Details on all active orders, including deadlines and required steps
- The progress status of any ongoing dishes
- Notifications about completed or canceled dishes

Write a single, clear paragraph that assigns roles and actions to each
    agent.
Your plan should optimize time by prioritizing urgent orders and
    minimizing movement overlap.

This planning routine is activated at the start and whenever an order is
    updated, completed, or canceled.
```

```
As the strategic planner in a kitchen with {total_num_agents} agents,
    your job is to design an efficient high-level plan to complete all
    active dish orders.

You are provided with:
- The spatial layout of the kitchen, including agents, tools, and
    ingredient locations
- A list of current orders, their components, and any timing constraints
- The current state of dishes being prepared
- A record of recently completed or canceled dishes

Compose a concise paragraph outlining how to allocate tasks and
    coordinate agents to complete the orders efficiently.
Your plan should account for timing, minimize idle time, and assign roles
    to avoid conflicts.

This planning step will run at initialization and every time there is a
    change to the set of orders.
```

```
You are a high-level planner for a kitchen with {total_num_agents} agents.

Your role is to generate a brief paragraph describing how agents should
    cooperate to complete dish orders efficiently.

You will receive:
- A detailed map of the kitchen, including all agent and item locations
- A list of active dish orders with deadlines and required steps
- The progress status of any ongoing preparations
- Updates about completed or canceled dishes

Create a clear, task-oriented plan that assigns responsibilities to
    agents, prioritizes dishes by urgency, and avoids agent overlap.

This planning is executed when the kitchen starts and whenever dish
    orders change.
```

## G  MODEL PRICES

In Table 6, we present the models' prices selected at current rates from their official APIs. OpenAI API price website: openai.com/api/pricing/, Anthropic API pricing: anthropic.com/pricing. For the case of Open-weights, we take the values from Llama-API provider: llama-api.com/pricing. All websites have been checked at date 25 March 2025.

Table 6: Model Prices from providers.

| Model | Company | Open Weights | Input Cost ($/M tokens) | Output Cost ($/M tokens) |
|---|---|---|---|---|
| claude-3.7 | Anthropic | ✗ | 3.00 | 15.00 |
| gpt-4o-v2 | OpenAI | ✗ | 2.50 | 10.00 |
| gpt-4o-mini | OpenAI | ✗ | 0.15 | 0.60 |
| Llama-3.1-70B-Instruct | Meta | ✓ | 0.80 | 2.80 |
| Qwen2.5-32B-Instruct | Alibaba | ✓ | 0.40 | 1.40 |

## H  TOKENS

Table 7: **Input token usage (in millions) across coordination strategies.** For Planner rows, values denote **Planner/Executor** token counts.

| | | Input Tokens (Millions) | | | | | |
|---|---|---|---|---|---|---|---|
| Models | | # Agents | | | | | |
| Planner/Orch. | Worker | 1 | 2 | 3 | 4 | 5 | 6 |
| *Individual* | | | | | | | |
| ✗ | GPT-4o-mini | 5.03 | 10.19 | 16.06 | 22.12 | 28.32 | 34.88 |
| ✗ | Llama-70B-Inst | 4.28 | 8.87 | 13.78 | 18.83 | 24.29 | 29.89 |
| ✗ | Qwen32B | 4.82 | 10.19 | 15.85 | 21.34 | 27.37 | 33.00 |
| ✗ | GPT-4o | 4.47 | 9.30 | 14.37 | 19.62 | 25.16 | 30.80 |
| ✗ | Claude-3.7 | 5.21 | 11.07 | 17.71 | 24.64 | 32.08 | 39.05 |
| *Orchestrator* | | | | | | | |
| GPT-4o | ✗ | 4.48 | 4.79 | 5.12 | 5.33 | 5.57 | 5.82 |
| Claude-3.7 | ✗ | 5.49 | 6.31 | 6.74 | 7.09 | 7.54 | 7.86 |
| *Planner (Planner / Executor)* | | | | | | | |
| GPT-4o | GPT-4o-mini | 0.73/2.01 | 0.94/4.06 | 0.99/6.46 | 0.53/4.35 | 0.88/8.98 | 1.17/14.73 |
| | Llama-70B-Inst | 0.93/2.19 | 0.98/4.52 | 1.05/6.92 | 1.07/9.35 | 1.11/11.90 | 1.15/14.56 |
| | Qwen32B | 1.01/2.20 | 0.99/4.57 | 1.01/6.89 | 1.07/9.53 | 1.03/12.03 | 1.10/14.68 |
| Claude-3.7 | GPT-4o-mini | 0.99/2.07 | 1.10/4.28 | 1.17/6.60 | 1.21/9.08 | 1.32/11.54 | 1.44/14.01 |
| | Llama-70B-Inst | 1.02/2.09 | 1.08/4.34 | 1.16/6.67 | 1.21/9.04 | 1.25/11.61 | 1.32/14.13 |
| | Qwen32B | 1.16/2.09 | 1.09/4.34 | 1.17/6.71 | 1.20/9.13 | 1.29/11.61 | 1.27/14.19 |

Table 8: **Output token usage (in thousands) across coordination strategies.** For Planner rows, values denote **Planner/Executor** output token counts.

| | | Output Tokens (Thousands) | | | | | |
|---|---|---|---|---|---|---|---|
| Models | | # Agents | | | | | |
| Planner/Orch. | Worker | 1 | 2 | 3 | 4 | 5 | 6 |
| *Individual* | | | | | | | |
| ✗ | GPT-4o-mini | 104.39 | 196.84 | 342.79 | 482.47 | 618.59 | 757.30 |
| ✗ | Llama-70B-Inst | 106.90 | 221.84 | 344.54 | 470.69 | 607.18 | 747.13 |
| ✗ | Qwen32B | 67.92 | 168.10 | 268.78 | 333.73 | 412.26 | 498.35 |
| ✗ | GPT-4o | 46.21 | 102.06 | 157.49 | 209.74 | 264.39 | 308.89 |
| ✗ | Claude-3.7 | 51.47 | 134.65 | 269.48 | 394.71 | 546.42 | 627.65 |
| *Orchestrator* | | | | | | | |
| GPT-4o | ✗ | 40.19 | 59.43 | 81.68 | 92.08 | 107.47 | 120.67 |
| Claude-3.7 | ✗ | 72.21 | 139.28 | 166.77 | 186.14 | 215.75 | 233.41 |
| *Planner (Planner / Executor)* | | | | | | | |
| GPT-4o | GPT-4o-mini | 35.65/5.49 | 46.93/10.67 | 57.34/15.92 | 22.10/10.22 | 32.76/19.65 | 43.38/29.09 |
| | Llama-70B-Inst | 32.31/6.70 | 35.26/12.67 | 37.30/18.33 | 36.21/23.49 | 36.89/29.01 | 36.97/33.31 |
| | Qwen32B | 33.81/6.80 | 36.47/12.85 | 33.44/19.10 | 38.26/24.78 | 34.97/29.75 | 36.76/34.35 |
| Claude-3.7 | GPT-4o-mini | 25.66/5.80 | 29.82/10.74 | 32.65/15.15 | 35.29/19.60 | 37.23/23.66 | 38.47/27.35 |
| | Llama-70B-Inst | 25.08/6.55 | 30.17/12.67 | 32.65/18.10 | 33.27/23.29 | 36.11/28.25 | 36.68/36.68 |
| | Qwen32B | 25.54/6.61 | 29.81/13.01 | 33.02/18.97 | 33.70/24.98 | 34.66/29.80 | 34.46/35.77 |

# I ABLATION STUDY

Results of the ablation study presented in Section 5. In this section, we present in Table 9 the results from the prompt ablations required to generate Table 3 presented in the paper.

Table 9: Results from Ablation Study: Concurrent Allocation in CuisineWorld showing the number of completed orders and associated costs for different models and methods.

| Models | | Completed Orders | | | | | | Cost ($) | | | | | |
|---|---|---|---|---|---|---|---|---|---|---|---|---|---|
| | | #Agents | | | | | | #Agents | | | | | |
| Plan./Orch. | Worker | 1 | 2 | 3 | 4 | 5 | 6 | 1 | 2 | 3 | 4 | 5 | 6 |
| *Orchestrator* | | | | | | | | | | | | | |
| Claude-3.7 | ✗ | 26 | 46 | 76 | 80 | 87 | 82 | 16.8 | 19.4 | 21.9 | 23.4 | 25.4 | 24.9 |
| *Planner* | | | | | | | | | | | | | |
| Claude-3.7 | GPT-4o-mini | 1 | 6 | 5 | 14 | 17 | 23 | 0.4 | 1.3 | 1.1 | 2.7 | 4.9 | 7.0 |

