# OpenReview forum: "Self-Resource Allocation in Multi-Agent LLM Systems"
_ICLR.cc/2026/Conference — Submitted to ICLR 2026_

### Official Review · Reviewer_LGck · 2025-10-25

**Soundness:** 2
**Presentation:** 1
**Contribution:** 2
**Rating:** 2
**Confidence:** 4

**Summary:**

This work tackles a key question for multi-agent AI systems: what's the best way to coordinate a team of LLM agents? The authors pit two strategies against each other in a simulated kitchen environment: a centralized 'Orchestrator' that dictates every agent's exact move, and a semi-decentralized 'Planner' that provides a high-level strategy, leaving individual agents to work out the details. The experiments reveal a crucial trade-off: while the Planner approach is far more cost-efficient (more tasks completed per dollar), the micro-managing Orchestrator consistently achieves higher absolute performance, completing more tasks overall. The paper argues for the Planner's efficiency, but its own data suggests this comes at the cost of peak performance.

**Strengths:**

1. As LLMs evolve into autonomous agents, exploring self-resource allocation is pertinent. The work builds on foundational ideas (for example Minsky's Society of Mind) and could inspire further studies on the applications.

2. The distinction between Individual, Orchestrator, and Planner methods in Experiment 2 provides a novel lens for analyzing centralized vs. decentralized coordination in LLM systems. The use of CuisineWorld as a dynamic benchmark is appropriate, and the efficiency metric (orders completed per dollar) offers a practical way to quantify cost-performance trade-offs, potentially influencing real-world deployments where computational costs are a barrier.

3. The findings on Planner efficiency and capability awareness may inform scalable multi-agent designs.

**Weaknesses:**

1. The experiment sets up a "strawman" comparison by evaluating out-of-the-box ability of LLM against the Hungarian Algorithm, a polynomial-time optimal solver for assignment problems. It is well-known that LLMs, especially without COT, perform poorly on strict combinatorial optimization tasks. The results "Figure 4 showing <20% accuracy even for GPT-4o" lead to misleading conclusions, suggesting the need for "more efficient alternatives" without truly assessing LLMs as orchestrators. A more meaningful setup would evaluate LLMs' ability to use tools like the Hungarian Algorithm for planning, rather than replacing it as a solver. Thus, Experiment 1 fails to support the paper's core claims.

2. The authors claim the Planner method "outperforms" the Orchestrator in handling concurrent actions, emphasizing improved efficiency. However, this conflates efficiency (orders per cost) with absolute performance (total orders completed). Table 2 data reveals the opposite: Orchestrators (Claude-3.7 with 98 orders for 6 agents) often achieve higher absolute performance than Planners (Claude-3.7 + Llama-70B with 77 orders). The Planner's efficiency gains come at the expense of task completion, representing a performance-cost trade-off rather than outright superiority. The abstract and conclusions misleadingly equate "higher efficiency" with "better," ignoring this trade-off and potentially misguiding readers.

3. The comparison between "On-the-fly Allocation" (where the planner must dynamically infer worker capabilities) and "Informed Allocation" (explicit capability hints) is poorly defined. The paper does not explain the inference mechanism—e.g., whether the planner observes failures, uses feedback loops, or has any basis for dynamic learning. Without such details, the finding that Informed Allocation performs better is obvious and unsurprising. The experiment offers no valuable insights into how LLMs could online-learn or infer agent capabilities, limiting its contribution.

**Questions:**

S: The paper is riddled with structural errors, such as duplicate "Problem Definition" sections on page 3 (labeled as both Section 3 and 4) with overlapping content.

Figure captions are erroneous (e.g., Figures 4 and 5 share identical captions, despite Figure 5 focusing on error analysis). Figure 9's layout is chaotic, misusing Figures 6–8 as subheadings, which hinders readability.

The Ablation Study (Table 3) is incomprehensible, with unexplained values like "+0.06" or "+2.43," rendering it valueless to readers. These issues reflect poor attention to detail and undermine the paper's credibility.

Q: How were paraphrased prompts generated, and why does the Planner show higher variability?

---

> ### Author Response · Authors · 2025-11-21
> **Experimental Design, Trade-offs, and Revisions**
>
> ## Comments on weaknesses
>
> **On Experiment 1: Resource Allocation**
>
> We thank the reviewer for raising this concern. The intention of Experiment 1 was not to claim that LLMs should replace polynomial-time optimal solvers such as the Hungarian Algorithm in certain scenarios.  Instead, the goal was to evaluate whether LLMs can reliably satisfy combinatorial constraints and maintain assignment validity when all task costs are explicitly given. This tests orchestration reliability, not optimality.
>
> If the LLM is provided with the answer from the Hungarian Algorithm, it will not need to generate any answer, it will only need to give it as the output. However, this task serves as a proxy to more complicated assignment tasks, such as the ones evaluated in Experiments 2 and 3.
>
> **Tradeoff between performance and efficiency**
>
> We agree that distinguishing efficiency (orders per dollar) from absolute task completion is important, and thank the reviewer for highlighting this. Our contribution is to analyze the cost–performance trade-off between orchestration strategies, not to claim that planners universally outperform orchestrators in all metrics.
>
> We revised the text to explicitly state:
> - The Orchestrator achieves the highest absolute task completion (as shown in Table 2)
> - The Planner achieves the highest efficiency across all worker-model configurations due to lower execution cost.
>
> **On-the-fly vs Informed Allocation**
>
> In Experiment 3, we want to evaluate how LLMs can allocate tasks to worker agents with different capabilities, and how much it can gain from lightweight capability priors.
>
> Concretely, in the *On-the-Fly* condition, the planner receives no explicit descriptions of worker abilities, similar to the previous experiment. The only information provided is the state from the environment (orders queue, order progress, task execution and workers status).
>
> In Contrast,  the *Informed Allocation* setting provides the planner with a small amount of structured prior knowledge: the cost-success rate of the models derived from previous results. The high level information is expected to provide the Planner LLM with information about which tasks should be allocated to the models with the goal of completing the maximum number of orders.
>
> The results show how a lightweight information can help a planner improve efficiency when compared to a scenario without information. We believe this result can help boost utilization and efficiency in an LLM-based multi-agent system. We have also modified some text of this experiment to clarify certain points based on your feedback.
>
> ## Comments on questions
>
> **Structural Errors**
>
> Thank you for pointing out these style errors and figure captions. We have revised the paper to amend the problems on Figures, header sections and more. We hope that the paper is now more clear.
>
> **On the Ablation study**
>
> Regarding the paraphrased prompts, we generated three semantically-equivalent paraphrases. These prompts are included in Appendix F. The purpose of the ablation study is to quantify the effect of the input prompts to the results. We ran the experiments on the strongest model setting on Orchestrator (Claude 3.7)  and Planner mode (Claude 3.7+GPT-4o mini). We show the results in Table 3 where with the difference to the results with the original experiment. We have made some change to make this more clear. Performance results are also shown in Appendix I.

---

> > ### Author Response · Authors · 2025-11-30
> >
> > Hello,
> >
> > We hope our response has helped clarify our work.
> > We remain available until the end of the rebuttal phase to address any additional comments and to further discuss your evaluation.
> >
> > Sincerely,
> > The authors

---

### Official Review · Reviewer_XMJS · 2025-10-26

**Soundness:** 3
**Presentation:** 2
**Contribution:** 3
**Rating:** 6
**Confidence:** 3

**Summary:**

This paper studies how multi-agent systems built upon LLMs can perform task and resource allocation autonomously. The authors compare three coordination strategies across multiple settings: (1) static assignment problems (benchmarked against the Hungarian algorithm), (2) concurrent multi-agent tasks in a simulated CuisineWorld environment, and (3) capability-aware allocation scenarios with both informed and on-the-fly capability estimation. The results show that larger LLMs generally yield higher assignment accuracy but at greater cost; the Planner setup outperforms the Orchestrator under concurrent execution by reducing idle actions; and explicitly encoding worker capabilities significantly improves overall efficiency, especially for weaker agents.

**Strengths:**

- The paper systematically studies three coordination paradigms across multiple domains, providing a multi-dimensional evaluation of LLM-based resource allocation.

- By explicitly encoding worker capabilities, the paper introduces a practical and generalizable way to improve multi-agent collaboration.

- The study systematically analyzes how the Planner adapts task distribution based on agents’ abilities, demonstrating that explicit capability information significantly improves overall system performance, especially when some agents are suboptimal.

- By directly comparing the Orchestrator and Planner strategies under concurrent action settings, the authors uncover clear efficiency advantages of the Planner.

**Weaknesses:**

- In the CuisineWorld concurrent-action experiments, comparisons are limited to Orchestrator and Planner paradigms. Including algorithmic or learning-based baselines (e.g., heuristic schedulers, RL-based task allocation, or rule-based controllers) would contextualize the performance improvements more convincingly.

- The reported cost metric does not clarify whether it includes token usage, API latency, or parallelization overhead. Presenting token-level statistics and latency–cost trade-offs would make the resource-efficiency analysis more interpretable.

- Although the authors analyze invalid assignments, the paper does not propose or evaluate mitigation methods. Adding a constraint-checking post-processor or structured prompt templates could reduce these failures and strengthen robustness.

- The paper does not report the number of trials, random seeds, or variance estimates. Confidence intervals or standard deviations should accompany results such as Completed Orders and Cost to support claims of performance differences.

- The accuracy of the assignment evaluation depends on GPT-4o acting as an automatic judge. This introduces potential bias, as both generation and evaluation rely on similar model families.

**Questions:**

- Both Section 3 and Section 4 share the same title, Problem Definition, which is likely a formatting or organizational oversight. Could you clarify whether these sections describe distinct problems? This duplication currently makes the paper structure confusing.

- What is the behavior of the Planner when the provided capability information is inaccurate or inconsistent? A robustness test under noisy capability descriptions would strengthen the claim that explicit ability modeling consistently improves allocation quality.

- How many assignment matrices were evaluated per model, and what was the distribution of task/agent counts? Are there scaling trends with matrix size?

---

> ### Author Response · Authors · 2025-11-21
>
> We thank the reviewer for the your comments and address the concerns and questions below.
>
>
> ## Comments on weaknesses
>
> **Contextualization**
>
> In the first experiment, we evaluate the ability of LLMs to perform task allocation by comparing their assignments to the optimal solution computed algorithmically. In the subsequent CuisineWorld experiments, our goal is to examine this ability under concurrent-action settings.
>
> Our primary goal in CusineWorld experiments is to isolate LLM-driven coordination behaviors under identical task specifications and observation models. Our primary goal in CusineWorld experiments is to isolate LLM-driven coordination behaviors under identical task specifications and observation models assume full state observability and do not operate through natural-language communication, making them incompatible for measuring efficiency using LLM-generated token costs.
>
> **Token-level statistics**
>
> The main tables in the paper report the computed LLM costs. Token-level statistics are provided in Appendix H, which includes the input/output token counts for all LLMs. Regarding the latency trade-offs, we do not consider them as they may depend on external factors such as internet connection, provider, or specific GPUs, whereas token counts and associated cost provide a stable, implementation-agnostic proxy for how much computing is used.
>
> **Mitigation methods**
>
> As mentioned in the paper, our experiments deliberately evaluate out-of-the-box LLM behavior without correction. Several lightweight mitigation techniques can directly address invalid assignments: post-hoc constraint checking (to enforce one-to-one matching), structured output formatting (e.g., JSON schema), and self-verification steps where the model recomputes the total cost. These interventions are simple to add, orthogonal to the allocation method, and could eliminate some errors identified in our analysis.
>
> **Statistics**
>
> In Experiment 1 (Resource Allocation), the evaluation is fully deterministic, it uses greedy decoding (temperature = 0) on a fixed set of 100 assignment matrices, yielding no run-to-run variance. For experiments 2 and 3 on CuisineWorld, each setting was executed once per agent configuration, which large number of evaluations that exhibit consistent trends.
>
> In order to asses the robustness of our results, we also ran an ablation study on the effect of the prompts. We generated 3 equivalent prompts and reran the strongest setting (Claude-3.7) for Orchestrator and Planner (with GPT-4o-mini). Results show that the main efficiency conclusions remain stable, with some higher variability for the Planner across agents due to the smaller executor model. In the three independent re-wordings, the Planner retains a clear efficiency lead. Results are documented in the paper.
>
>
> **Judge bias**
>
> As indicated in the paper, GPT-4o is used as a judge only in Experiment 1 (Resource Allocation). The judge simply compares the model’s output to the ground-truth assignment, allowing free-form LLM generation while using a consistent verification mechanism. It is been shown that some LLMs have an answer for certain texts, but in this case the judge only needs to match the generated assignment to the exact ground truth, which substantially limits the potential for biased evaluation.
>
> ## Responses to questions
>
> **Duplicated section titles**
>
> We acknowledge the confusion caused by Sections 3 and 4 sharing the title Problem Definition. This has been corrected in the revised version, and we believe the updated structure is now clear.
>
> **Robustness**
>
> In the third experiment, we evaluate the Planner under two conditions: *on-the-fly* and *informed allocation*. In the informed setting, the Planner is given explicit ability information for each worker model, derived from their performance in earlier experiments. When the Planner must infer abilities itself, performance generally decreases.
>
> We did not evaluate scenarios where the Planner is intentionally given incorrect capability information. But we do not foresee a use case where we want to deliberately confuse the planner model, since all models are controlled by the same entity. However, based on the reduced performance observed in the on-the-fly setting, where inferred capabilities can be imperfect, where inferred capabilities can be imperfect. We did evaluate the effect of varying prompt templates in the ablation study, included in the results section and Table 3, which supports the robustness of the evaluation to prompt variations.
>
> **Assignment matrices**
>
> In Experiment 1, we evaluated a total of 100 matrices, each of size 6×6, using model temperature = 0 to ensure deterministic behavior. An example assignment problem is provided in Appendix A.

---

> > ### Comment · Reviewer_XMJS · 2025-11-26
> >
> > Thanks for the rebuttal, most of my concerns are addressed. I will keep my positive score.

---

> > > ### Author Response · Authors · 2025-11-27
> > >
> > > Thank you for your comments and your positive assessment of our submission. We appreciate the time committed to reviewing our work and we believe your comments have helped us better understand the strengths and weaknesses of our paper.
> > >
> > > If there are any remaining points you feel we should address, we would be glad to incorporate them. Please feel free to let us know if any additional clarification would further strengthen the paper.

---

### Official Review · Reviewer_GPze · 2025-10-31

**Soundness:** 2
**Presentation:** 2
**Contribution:** 2
**Rating:** 2
**Confidence:** 3

**Summary:**

This paper explores the ability of LLMs to allocate tasks and resources in multi-agent systems, comparing three coordination strategies: Individual, Orchestrator, and Planner. The authors evaluate these approaches in three settings: (1) a static assignment problem with explicit costs; (2) a dynamic cooking environment (CuisineWorld) with delayed rewards; and (3) a capability-aware setting with heterogeneous agents. The key findings include: (i) LLMs can solve assignment problems, with accuracy improving with model size—at significant computational cost; (ii) the Planner paradigm achieves higher efficiency and better agent utilization than the Orchestrator in concurrent tasks; and (iii) while LLMs struggle to infer agent capabilities on the fly, their allocation quality improves markedly when provided explicit capability information, highlighting their sensitivity to worker heterogeneity.

**Strengths:**

1.	The topic addressed in the paper is important: enabling LLM-based agents to perform self-resource allocation in multi-agent systems, which is critical for advancing the field of autonomous agents.
2.	The findings are intuitive and practically relevant, for example, the planner approach outperforms the orchestrator, and that explicitly providing information about worker capabilities improves allocation strategies.

**Weaknesses:**

1.	Insufficient analysis: While the experiments provide empirical evidence, the paper falls short in explaining the underlying reasons behind the observed results. Merely presenting factual outcomes without deeper interpretation limits the contribution.
2.	Experimental setup limitations: It would be better to include tests with state-of-the-art models such as GPT-5, Claude-4.5 Sonnet, or Qwen3. And the evaluation relies solely on toy environments (e.g., CuisineWorld). Demonstrating applicability in real-world scenarios will be more exciting.
3.	There are some typos in the paper:
a)	There are incorrect citation formats, I highly encourage the author go through the paper for a double check.

b)	Sectioin 3 and 4 with duplicated tiltle

**Questions:**

see weakness

---

> ### Author Response · Authors · 2025-11-21
> **Experimental Design, Analysis Intent, and Revisions**
>
> Thank you for your comments and feedback. We appreciate the time you put into reviewing our work. We believe that our paper makes a significant contribution to the field of autonomous agents by evaluating their
>
> **Experimental setup**
>
> Our choice of models used relies on the models accessible through our platform and the limits of our budget. We believe the current set represents a broad range of strong and smaller models from different companies (e.g. OpenAI, Anthropic, Meta, Alibaba).
>
> Regarding the experimental environments, we include
> - Experiment 1: Assignment Problem, where we created a controlled evaluation of the assignment problem on our 6x6 assignment matrices, paired with ground truth solution from the Hungarian Algorithm.
> - For the subsequent experiments, we rely on the CuisineWorld benchmark, which allows us to execute the experiments on our 3 evaluated strategies while containing desirable features such as concurrency and subtasks.
>
> We considered adding the results from other environments, mainly Coherent [1], but other environments differ substantially in their task/sub-tasks division or agent behavior. This makes direct comparison across the three coordination strategies less interpretable.
>
> **Analysis**
>
> The aim of this paper is not to offer a theory for every behavior we observe, but to provide a careful, empirical look at how different LLM-based coordination approaches (Individual, Orchestrator, and Planner) respond to changes in task structure, reward timing, and agent capabilities. By isolating these factors, we show how they influence allocation quality, efficiency, and agent usage across the three experimental settings. Our analysis highlights where errors arise, how performance scales with model size, how planners and executors interact, and how systems react to differing worker abilities. Taken together, these results support our main conclusion: the Planner approach consistently yields more efficient task execution and better use of resources than centralized orchestration. Our goal is to offer a clear, reproducible foundation for studying LLM-driven self-allocation, an area where systematic baselines and diagnostics are still limited.
>
> **Typos and incorrect citations**
>
> Thank you for pointing out some style errors and typos. We have now revised the paper in order to to correct citation formatting, remove duplicated section headers, and improve overall readability.
>
> **Overall clarification**
>
> We would like to emphasize some contributions our work provides:
>
> - A systematic comparison of three LLM-based coordination paradigms (Individual, Orchestrator, Planner).
> - A capability-aware benchmarking protocol, enabling controlled analysis of worker heterogeneity, which prior work lacks.
> - Practical insights into scaling behavior, concurrency, and capability conditioning, which we believe are valuable for building efficient LLM-driven multi-agent systems.
>
> We hope these clarifications address the concerns raised.
>
>
> ---
>
> [1] Liu, Kehui, et al. "Coherent: Collaboration of heterogeneous multi-robot system with large language models." 2025 IEEE International Conference on Robotics and Automation (ICRA). IEEE, 2025.

---

> ### Author Response · Authors · 2025-11-30
>
> Hello,
>
> We hope our response helped solving some of your concerns about our work.
> We remain available to solve some of your remaining concerns and help increase the score of your evaluation.
>
> Sincerely,
> The authors

---

### Official Review · Reviewer_1SoX · 2025-11-01

**Soundness:** 3
**Presentation:** 3
**Contribution:** 3
**Rating:** 6
**Confidence:** 4

**Summary:**

This paper investigates self-resource allocation in multi-agent LLM systems, formalizing a budget-aware objective and comparing three organizational regimes—Individual (decentralized), Orchestrator (centralized, fine-grained action emission), and Planner (centralized high-level planning with low-cost decentralized executors). The authors evaluate (i) static assignment, where LLMs attempt to reproduce Hungarian-optimal matchings; (ii) concurrent scheduling in a simulated task environment, measuring throughput-per-dollar and idle-action rates; and (iii) capability-aware allocation, contrasting on-the-fly skill inference with explicit ability hints. Across settings, the Planner consistently yields the best cost-efficiency and fewer no-ops, while exposing coarse worker capability profiles further boosts performance in heterogeneous pools; by contrast, direct LLM orchestration remains error-prone on constrained matching (e.g., miscounted costs, constraint violations). Overall, the work contributes a reproducible evaluation protocol and practical guidance: concentrate expensive reasoning in compact planning steps, delegate execution to cheaper models, and surface lightweight ability profiles to the planner under budget constraints.

**Strengths:**

- By casting self-resource allocation as a cost-aware comparison of Individual / Orchestrator / Planner regimes, the paper adds a fresh systems lens beyond prior role-playing multi-agent frameworks.
- The methodology pairs static assignment with a cooperative CuisineWorld/Overcooked-style concurrent benchmark, yielding both groundable checks and ecologically valid coordination stress-tests.
- The three regimes are distinguished crisply, and the Planner is explained via a ReAct-like separation of reasoning and action, complemented by interpretable metrics (throughput-per-dollar, idle/no-op).
- The results translate into actionable guidance for real systems—concentrate expensive reasoning in compact planning and delegate execution to cheaper agents.

**Weaknesses:**

- The experiments are currently restricted to a single benchmark, CuisineWorld, which does not guarantee that the observed behaviors generalize to other domains such as web-based agents tasks.

- Several key tables and figures present point estimates without reporting variance, confidence intervals, or statistical significance across multiple runs. Without such analysis, it remains unclear whether the reported efficiency gains are consistent, reproducible, or simply artifacts of stochastic sampling and prompt variability.

- The replanning mechanism—including trigger frequency, cost thresholds, and congestion conditions—is not systematically ablated or visualized, leaving open when and why replanning contributes most to performance improvements.

- The evaluation stops at six agents and moderate task density, offering limited insight into how the proposed Planner and Orchestrator architectures scale under larger or more heavily coupled systems.

**Questions:**

- Could the authors rerun the Planner–Orchestrator comparison under a fixed budget or total token constraint to isolate architecture effects from pricing differences?

- If the Orchestrator adopted a hierarchical two-level scheme (strong model planning, cheap model executing), would its efficiency approach that of the Planner?

- Code, prompts, and data be fully open-sourced?

---

> ### Author Response · Authors · 2025-11-21
> **Generalization, Reproducibility, and Scaling**
>
> ## Comments on weaknesses
>
> **Generalization beyond CuisineWorld**
>
> Our current submission focuses on CuisineWorld because it offers (i) controllable multi-agent concurrency, (ii) grounding for static assignment evaluation, and (iii) interpretable cost and latency modeling. While the experiments use this benchmark, the method itself is domain-agnostic: the Planner relies on symbolic action graphs and delegation interfaces that do not assume simulator-specific affordances.
>
> We also considered including an additional benchmark (e.g., COHERENT [1]). However, such environments would not allow a direct comparison of results across benchmarks, making true “apple-to-apple” evaluation difficult. Adding them would therefore expand the scope without improving comparability.
>
> **Reproducibility and variance**
>
> For Experiment 1 (Resource Allocation), evaluation is executed in a deterministic method because we use greedy decoding (temperature = 0).
>
> For the subsequent experiments, each setting was executed once per agent configuration, which  large number of evaluations that exhibit consistent trends.
>
> To assess variability, we ran an ablation study on the effect of the prompt. We generated 3 equivalent prompts and reran the strongest setting (Claude-3.7) for Orchestrator and Planner (with GPT-4o-mini). Results show that the main efficiency conclusions remain stable, with some higher variance for the Planner due to the smaller executor model. Across three independent re-wordings, the Planner retains a clear efficiency lead. Results for the efficiency table included below:
>
> ### Efficiency (orders / $)
>
> #### (a) Orchestrator – Claude-3.7
>  | Variant| 1| 2| 3| 4| 5| 6|
> |-|-|-|-|-|-|-|
>  | Original| 1.54 | 2.38 | 3.47 | 3.42 | 3.42 | 3.29 |
>  | Paraphrase (avg.) | 1.49 | 2.33 | 2.90 | 3.53 | 3.29 | 3.62 |
>  | Δ (Orig − Para) | 0.06 | 0.04 | 0.56 | 0.11 | 0.130 | 0.33 |
>
> #### (b) Planner – Claude-3.7 + GPT-4o-mini
>  | Variant| 1| 2| 3| 4| 5| 6|
>  |-|-|-|-|-|-|-|
>  | Original| 2.45 | 4.48 | 4.47 | 5.12 | 3.48 | 3.29 |
>  | Paraphrase (avg.) | 2.07 | 2.05 | 3.40 | 3.61 | 3.51 | 4.38 |
>  | Δ (Orig − Para) | 0.38 | 2.43 | 1.06 | 1.50 | 0.03 | 1.08 |
>
> **Replanning mechanism ablation**
>
> Currently, the replanning mechanism is event-driven, the planner is called every time a new order is introduced into the queue, completed, or removed. This design choice ensures that the Planner only performs high-cost reasoning when there is a significant change in the environment.
>
> **Limited scaling**
>
> Our choice of n ≤ 6 agents is constrained by the CuisineWorld environment. Beyond this number, the number of agents leads to unnecessary overhead and task parallelism collapses due to limited workspace and single-use appliances. The original MindAgent paper [2] also evaluates up to six agents and notes similar limitations regarding realism and interpretability. A related Overcooked-based benchmark, Collab-Overcooked [3], similarly reports increased complexity and cost as the number of agents grows.
>
> ## Responses to questions
>
> **Fixed budget**
>
> Initially, we considered the approach of a fixed budget. However, we noticed it introduces several issues. If the budget is below the threshold required for each method to reach its natural “maximum score,” then results reduce to comparing which method completes more steps under this budget constraint.  We must also decide where the budget should be set (e.g., 25% / 50% / 75% of the unconstrained requirement), which introduces another hyperparameter. If the budget is above the threshold, then it does not influence the results and become the same as our current evaluation.
>
> **Orchestrator with a hierarchical two-level scheme**
>
> The Planner already embodies a hierarchical approach: a strong model performs high-level planning, and cheaper models execute low-level actions.
>
> The orchestrator embraces a centralized approach. If the orchestrator is also made a 2-level approach, it would not be more efficient than the Planner, as it would be calling the strong model for every action and letting the smaller models execute them which would generate an unnecessary extra step and lose in efficiency when compared to the Planner. Such a hierarchical orchestrator would not surpass the Planner in cost-efficiency.
>
> **Open-source**
>
> Prompts are available in the appendices, and code is included in the supplementary materials. We commit to releasing all code, prompts, and data online as open-source for the camera-ready version.
>
> ---
>
> [1] Liu, Kehui, et al. "Coherent: Collaboration of heterogeneous multi-robot system with large language models." 2025 IEEE International Conference on Robotics and Automation (ICRA). IEEE, 2025.
> [2] Gong, Ran, et al. "Mindagent: Emergent gaming interaction." Findings of the Association for Computational Linguistics: NAACL 2024. 2024.
> [3] Sun, Haochen, et al. "Collab-Overcooked: Benchmarking and evaluating large language models as collaborative agents." arXiv preprint arXiv:2502.20073 (2025).

---

> > ### Author Response · Authors · 2025-11-30
> >
> > Hello,
> >
> > Thank you for your comments and your review of our work.
> > We hope we have solved some of your concerns in our response.
> > We remain available to answer any more comments and solve your remaining concerns.

---

### Meta-Review · Area_Chair_84TE · 2025-12-14

**Summary:**

The major concerns are about the technical depth, including insufficient analysis about the techniques and the performance improvement, lack of details and analysis about the inference mechanisms, and the experiment settings (including the selected task). After the rebuttal, most concerns still remain. We encourage the authors to take the reviews seriously into consideration when they revise the paper.

**Reviewer Concerns:**

Most concerns are not fully addressed. More time and efforts (than a rebuttal) are needed to address these concerns.

**Reviewer Scores:**

The scores are reasonable.

---

### Decision · Program_Chairs · 2026-01-26

Reject